# An Evaluation of Oral Anticoagulant Safety Indicators by England’s Community Pharmacies

**DOI:** 10.3390/pharmacy12050134

**Published:** 2024-08-29

**Authors:** Sejal Parekh, Lingqian Xu, Carina Livingstone

**Affiliations:** 1Primary Care Strategy and NHS Contracts Group, Primary, Community and Personalised Care Directorate, NHS England, London SE1 8UG, UK; 2Pharmacy, Optometry and Dental Strategic Analysis Group, Primary, Community and Personalised Care Directorate, NHS England, London SE1 8UG, UK; jessie.xu@nhs.net; 3NHS Specialist Pharmacy Services, The Causeway, Ground Floor, Worthing BN12 6BT, UK

**Keywords:** anticoagulants, medicine safety, community pharmacy, pharmacists, pharmacy quality scheme, primary care

## Abstract

Background: Anticoagulants are life-saving medicines that can prevent strokes for patients diagnosed with atrial fibrillation (AF) as well as treating patients with venous thromboembolism (VTE), but when used incorrectly, they are frequently associated with patient harm. Aim: To evaluate the impact of community pharmacy teams on optimising patient knowledge and awareness and improving medication safety from the use of anticoagulants. Methods: Two national audits, consisting of 17 questions assessing and improving patients’ understanding of anticoagulant therapy, identifying high-risk patients, and contacting prescribers when clinically appropriate were incentivised for England’s community pharmacies in 2021–2022 and 2023–2024 using the Pharmacy Quality Scheme (PQS) commissioned by NHS England. Results: Approximately 11,000 community pharmacies audited just under a quarter of a million patients in total, whilst making almost 150,000 interventions for patients taking oral anticoagulants, i.e., identifying and addressing medication issues which could increase the risk of bleeding/harm. Out of the 111,195 patients audited in 2021–2022, only 24,545 (23%) patients were prescribed vitamin K antagonists. The remaining patients were prescribed direct oral anticoagulants (DOACs). By 2023–2024, this decreased to 17,043 (16%) patients. Most patients knew that they were prescribed an anticoagulant (95.6%, 106,255 in 2021–2022 and 96.5%, 101,006 in 2023–2024, *p* < 0.001). Discussion: The audits resulted in a statistically significant increase in patients with a standard yellow anticoagulant alert card, as identified in audit 2 (73,901 66.5% in 2021–2022 to 76,735, 73.3% in 2023–2024, *p* < 0.001). Furthermore, fewer patients were prescribed concurrent antiplatelets with an anticoagulant (6021; 4.6% in 2021–2022 to 4975; 4% in 2023–2024, *p* < 0.001). Although there was an increase in the number of patients prescribed NSAIDs with anticoagulants, more of these patients were also prescribed gastroprotection concurrently (927 77.2% in 2021–2022 to 1457 84.1% in 2023–2024, *p* < 0.05). The majority of patients on warfarin had their blood checked within 12 weeks. Further there was an increase for these patients in the percentage of people prescribed VKAs who knew dietary changes can affect their anticoagulant medicine (16,764 67.4% in 2021–2022 to 12,594 73.9% in 2023–2024 *p* < 0.001). Conclusions: Community pharmacy teams are well placed in educating and counselling patients on the safe use of anticoagulants and ensuring that all patients are correctly monitored.

## 1. Introduction

Anticoagulants are life-saving medicines that can prevent strokes for patients diagnosed with atrial fibrillation (AF) as well as treating patients with venous thromboembolism (VTE) [1]. However, this class of medicine is known to be high risk as they are frequently associated with significant patient harm when used incorrectly, i.e., if not prescribed correctly and/or taken in accordance with correct instructions [2].

A patient safety alert by the UK’s National Patient Safety Agency (NPSA) highlighted 15 high-risk factors correlating to safety incidents from anticoagulants [3]. These incidences comprised 8–10% of all preventable medication-associated hospital admissions [3]. Anticoagulant medicines are known to be a frequent cause of preventable and avoidable harm which that have resulted in hospital admissions.

In 2017, the World Health Organisation launched the third Global Patient Safety Challenge: Medication Without Harm for high-risk drugs to reduce harm by 50% in the following 5 years. Anticoagulants carry a risk of bleeding if the dose is too high and a risk of thrombosis if it is too low [4]. The NHS Business Services Authority (NHSBSA) developed a set of medication safety-prescribing indicators to monitor and reduce associated medication errors [4].

In 2020/21, the medicines safety dashboards indicated that over 100 patients were admitted to hospital with gastrointestinal bleeding (GI) because they were concurrently prescribed an oral anticoagulant with a Non-Steroidal Anti-Inflammatory Drug (NSAID). A further 327 patients were admitted with GI bleeding as a result of being concurrently prescribed an oral anticoagulant and an antiplatelet. Neither of these patient groups were prescribed gastroprotective medication to reduce this risk of GI bleeds [5].

From 2014 to 2023, the number of anticoagulants prescribed almost doubled, whilst drug expenditures increased 10-fold in England [6]. In 2023, 20,930,879 anticoagulant prescriptions were dispensed, with actual medicine expenditures accounting for £755,412,462.25 [6].

These trends indicated that prescribers were moving away from warfarin, preferring DOACs instead [6]. In 2020, during the coronavirus pandemic, a guidance for the management of anticoagulant services was released, addressing face-to-face services and anticoagulant prescribing [7,8]. Where anticoagulation was indicated, the guidance recommended that patients prescribed warfarin be switched to a DOAC. The aim was to reduce the monitoring requirements, as warfarin requires regular International Normalised Ratio of Clotting (INR) monitoring to calculate and adjust dosing. In contrast, patients prescribed DOACs receive anticoagulant therapy without requiring frequent blood monitoring, which would otherwise pose a potential COVID-19 transmission risk. Supporting both healthcare professionals’ and patients’ was even more significant for managing changes in monitoring and concordance with anticoagulant use.

However, as a result of these recommendations, there have been incidents of patients taking both warfarin and DOACs concurrently, which can result in “over anticoagulation” and a subsequent increased risk of bleeding [8]. To reduce this risk, the Medicines and Healthcare Products Regulatory Agency (MHRA) recommended that healthcare professionals should ensure that warfarin treatment is stopped prior to starting a DOAC and encourage patients to return any medication no longer needed to community pharmacies [9].

Community pharmacy teams are therefore in the right position to promote the of anticoagulants safely and to ensure a safe transition from warfarin to DOACs, where they provide face to face care and counselling as well as review patient-held yellow booklets [10]. Whilst dispensing and supplying medicines to patients, they can make interventions such as identifying any modifications in dosage to support optimisation, managing side effects or drug interactions [11,12].

Drug interactions that can lead to serious bleeding and/or thrombosis and can be caused by commonly prescribed medicines or medicines that can be purchased over the counter from pharmacies. For this reason, it is essential that patients taking anticoagulants always remind prescribers of their anticoagulant medication so that they can assess the risk and take advice from a healthcare professional before taking over-the-counter medication. Patients should routinely be provided with a standard yellow anticoagulant alert card to remind them of this risk in accordance with the alert issued by the MHRA.

This study aims to evaluate the impact of community pharmacy teams on optimising patient knowledge and awareness and improving medication safety from the use of oral anticoagulants.

## 2. Materials and Methods

### 2.1. Study Design

Two audits were incentivised for England’s community pharmacies in 2021–2022 and 2023–2024 under the Community Pharmacy Contractual Framework and the Pharmacy Quality Scheme (PQS) commissioned by NHS England [13]. These audits were built on a voluntary audit conducted in 2017/2018, was aimed at determining patient awareness of key information about anticoagulants [10].

The audit questionnaire consisted of 17 questions assessing and improving patients’ understanding of anticoagulant therapy, identifying high-risk patients, and contacting prescribers when clinically appropriate (Appendix A). The data collection tool included patient demographics; the name, dose and duration of anticoagulant prescribed; any concomitant medicines prescribed; whether gastroprotection was prescribed; conversations/contact with patients to support understanding of their medication; and whether the patient was referred to their general practitioner (GP) for a clinical review for suitable gastroprotection. A free-text option was available for general comments and additional information community pharmacy teams wanted to provide. The audit standards were applied to both audits and have been listed in the interim report and allowed a baseline to be measured [14].

### 2.2. Setting and Participants

The study population was any patient aged 18 and above presenting at community pharmacies in England with a prescription for any oral anticoagulant (a vitamin K antagonist (VKA), such as warfarin, acenocoumarol or phenidione), or a DOAC, such as apixaban, dabigratran, edoxaban or rivaroxaban)

All community pharmacy contractors in England are eligible to participate in this voluntary incentivised scheme and were invited to participate. Pharmacy contractors were asked to collect data for 2 weeks to reach a minimum sample size of 15 patients. If they were unable to achieve 15 patients over the 2 weeks, the audit was extended for a further 2 weeks to try to achieve the minimum sample size. If at this point, they were unable to achieve the sample size of 15 patients, they could submit the data. In addition, community pharmacy teams were required to follow up with patients they referred to prescribers to understand what actions were taken.

### 2.3. Data Collection

The data were collected by community pharmacy teams for the duration of the PQS. (In England, community pharmacy teams can comprise of pharmacists, registered pharmacy technicians, trainee pharmacists, trainee pharmacy technicians, dispensary staff, and medicine counter assistants). Audit 1 was conducted between 1 September 2021 and 31 March 2022. For audit 2, minor revisions were made to the data collection tool reflecting the recommendations and anomalies found from the analysis of audit 1 (e.g., the age range was amended to patients over 18 only). Pharmacy teams conducted audit 2 between 1 June 2023 and 31 March 2024.

The data were collected using a snap survey (which is a software programme for questionnaire design publication data collection, analysis and reporting for all modes of survey research that NHSA BSA use), which was accessed via the Manage Your Service (MYS) application. The data from audit 1 were reviewed to produce an interim report with recommendations to support quality and safety improvement for the reaudit in 2023–2024 (audit 2) [14].

Prior to participating in the 2023/24 PQS audit, community pharmacy teams were required to have implemented into their day-to-day practice, the findings and recommendations from audit 1 [14]

### 2.4. Data Analysis

Data were extracted from the NHSBSA MYS portal following submission from community pharmacy contractors. The data was analysed descriptively and using the chi-square test of independence comparing the equality of proportions was used to measure the differences between the standards over time using R statistical software version 4.2.1 (R Foundation for Statistical Computing). The results are reported as frequencies (and percentages) and *p* values for differences between the interventions over time.

## 3. Results

A total of 245,719 patients were audited, with 6605 community pharmacies participating in both audits and 10,899 pharmacies participating in either of the two audits. The majority of the data for both audits were collected when the patient attended the pharmacy to collect their medication 49.5% (64,991) in 2021–2022 vs. 55.3% (69,354) in 2023–2024; conversation with the patient by telephone 34.9% (45,809) in 2021–2022 vs. 28.0% (35,074) in 2023–2024 (Figure 1).

Representatives collected medication on behalf of patients for 7.9% (10,349) in 2021–2022 and 8.6% (10,843) in 2023–2024 (Figure 1). The average age of the patients analysed was 72.6 years for PQS 2011–2022 and 72.3 years for PQS 2023–2024.

The distribution of data from the regions is the same for both audit years, with the Midlands collecting the most patient data, followed by Northeast and Yorkshire, and Northwest Regions, where the most patient data were collected (Appendix A). Some data entries contained blank data as representatives collected medication for patients, and they did not always know patient details/could not complete the audit for the patient. Therefore, complete data were included in the final analysis. Incomplete data entries were excluded. The final analysis consisted of 111,195 patients from 2021–2022 (audit 1) and 104,677 from 2023–2024 (audit 2).

Table 1 shows the results against the standards for Audits 1 and 2. For standard 1, there has been a small percentage increase in the knowledge of key information, where the majority of patients knew they were taking an anticoagulant (106,255; 95.6% in 2020–2021 to 101,006; 96.5% in 2023–2024 *p* < 0.001); 4 out of every 5 patients knew about the symptoms of over anticoagulation (89,171; 80.2% in 2020–2021 vs. 85,071; 81.3% in 2023–2024 *p* < 0.001).

In addition, a greater percentage of patients understood the symptoms of anticoagulation, from 76.5% in 85,029 to 80.2% in 83,902 (Table 2). A greater percentage of patients were aware that they needed to receive professional advice before buying OTC remedies, herbal products, or supplements (80.2%; 89,171) in 2021–2022 to (81.3% 85,071 *p*< 0.001) in 2023–2024. Finally, for those prescribed a VKA, there was an increase in patients who knew dietary changes can affect their anticoagulant medicine (16,764 67.4% in 2021–2022 to 12,594 73.9% in 2023–2024 *p* < 0.001).

For standard 2, there was a significant increase from 66.5% of patients (in 2021–2022) to 73.3% (in 2023–2024) having a standard yellow anticoagulant alert card. There was a small increased in the number of patients offered a standard yellow anticoagulant alert card if they were identified as not having one—96.5% (25,459) in 2020–2021 to 94.6% (27,942) in 2023–2024 (Table 3).

For standard 3, fewer patients were prescribed antiplatelets concurrently 6021; 4.6% in 2021–2022 vs. 4975; 4.0% in 2023–2024; *p* value < 0.001. Furthermore, when these patients were concurrently prescribed an antiplatelet agent, a greater percentage were also prescribed gastroprotection—5273 (87.6%) in 2021–2022 and 4428 (89%) in 2023–2024 (Table 4). More than half of the patients who were not prescribed gastroprotection were referred to their GPs for a review; 217 (29.0%) patients in 2021–2022 and 151 (27.6) patients in 2023–2024 patients were prescribed gastroprotection as a result of the intervention (Table 4).

For Standard 4, there was an increase in the number of patients concomitantly prescribed an NSAID and an anticoagulant from 1201 (0.9%) in 2021–2022 to 1732 (1.4%) in 2023–2024, which led to community pharmacy teams intervening and contacting the prescriber to review the patient (Table 5). There was also an increase in the number of patients co-prescribed an NSAID, an anticoagulant and gastroprotection from 927 (77.2%) in 2021–2022 to 1457 (84.1%) in 2023–2024 (Table 5).

The majority of patients prescribed warfarin had their blood checked within 12 weeks for both audits—18,446 (99.3%) in 2021–2022 and 11,730 (99.1%) in 2023–2024. When an INR blood test was not completed within 12 weeks, community pharmacy teams investigated why and made arrangements for the INR to be completed in 42 of 138 patients in 2021–2022 and 21 of 102 patients in 2023–2024 (Table 6).

## 4. Discussion

A total of 245,719 patients were audited, with 6605 community pharmacies participating in both audits and 10,899 pharmacies participating in either of the two audits. Out of the 111,195 patients in 2021–2022, only 24,545 (23%) patients were prescribed a VKA. The remaining patients were prescribed DOACs. By 2023–2024, the number of patients audited and prescribed a VKA had further decreased to 17,043 (16%) patients. This audit shows there has been a change in the prescribing of anticoagulants compared to previous UK studies where warfarin was the most frequently prescribed anticoagulant [8,10]. More patients were aware they were prescribed an anticoagulant (95.6%, 106,255) in 2021–2022 and 96.5% (101,006) in 2023–2024. There was a statistically significant increase from 66.5% in 2021–2022 to 73.3% in 2023–2024, *p* < 0.001 having a standard yellow anticoagulant alert card.

### 4.1. Direct Oral Anticoagulants vs. Vitamin K Antagonists

Out of the 111,195 patients in 2021–2022, only 24,545 (23%) patients were taking VKA. By 2023–2024, this had further decreased to 17,043 (16%) patients. The remaining patients were prescribed DOACs. Historically, warfarin has dominated the choice of oral anticoagulation treatment [10]. DOACs have several benefits compared to VKAs, specifically warfarin, which is notorious for medication interactions [15]. In England, this initial change has occurred as a result of the COVID-19 pandemic and obvious infection control measures [9]. Several RCTs have established the non-inferiority of the anticoagulation qualities of DOACs compared with those of warfarin [16,17,18]. The switch in these patients has several advantages. First, regular INR monitoring is no longer required—the mechanism of action of DOACs does not alter the INR, and therefore, less frequent drug safety monitoring (e.g., renal function) is required [8]. Additionally, strict dietary requirements are not necessary for DOACs, which can be tedious for patients. DOACs interact with fewer medicines than warfarin (which has 238 interactions, whilst apixaban has 130 interactions, dabigatran has 152 interactions and edoxaban has 140 interactions) [15]. This is advantageous as anticoagulants are often taken lifelong and patients will undoubtedly require other medications at various time points. Collectively, the change in the class of drugs has resulted in a theoretical risk reduction in medication related errors and patient safety-associated risks from what was historically a single treatment option with the use of warfarin. DOACs still interact with many medications, but this is significantly lower than warfarin. The impact of the national change of anticoagulant prescribing is beneficial to both the patient in terms of no longer having to adhere to strict diets or requiring frequent blood monitoring and ongoing dose adjustment as well as economically, with less resources required to manage these patients (albeit acknowledging that fewer patients are prescribed VKAs in preference to DOACs than ever before). However, DOACs are dosed against renal function, which, if not checked regularly, may result in patients not receiving the corrected recommended dose and potentially being either anticoagulated or over anticoagulated [19].

### 4.2. Patient Awareness and Education

Most patients on anticoagulants were aware that they were being prescribed anticoagulants. These audits were carried out for just under 250,000 patients in England over a 3-year period and showed little change in knowledge despite the change in the class of anticoagulants. These results correspond to those of similar studies in England [10,20]. However, these findings oppose international data, as similar studies have indicated that many patients taking anticoagulants require more education [20,21,22].This difference may be due to a national push for GPs to start patients on DOACs (specifically Edoxaban) as well as switching patients where clinical appropriately [23]. This has led to a coordinated response in educating patients on the use of oral anticoagulants over a significant time period. The audits resulted in a statistically significant increase in patients with a standard yellow anticoagulant alert card, as identified in audit 2 (73,901 66.5% in 2021–2022 to 76,735, 73.3% *p* < 0.001 in 2023–2024). The majority of patients on a VKA (warfarin) had their blood checked within 12 weeks 18,446 (99.3%) in 2021–2022 and 11,730 (99.1%) in 2023–2024 and knew that dietary changes can affect their anticoagulant medicine (95.6%, 106,255 in 2021–2022 and 96.5%, 101,006 in 2023–2024, *p* < 0.001).

### 4.3. Medication Safety and the Role of Community Pharmacy Teams

Community pharmacy teams were proactive in counselling and educating patients about their medication. Over the course of the two audits, pharmacists and their teams provided 148,376 interventions to the 245,719 patients involved. Moreover, embedding the recommendations of the first audit significantly improved safety over time between audits. They offered key advice to patients regarding their indication for being prescribed an anticoagulant (4607 times in 2021–2022 and 3478 times in 2023–2024); symptoms of over anticoagulation e.g., unexplained bruising and nose bleeds (25,478 times in 2021–2022 and 20,308 times in 2023–2024); reminders to check with an appropriate healthcare professional before taking OTC medicines, herbal products, or supplements (21,033 times in 2021–2022 and 18,889 times in 2023–2024). In accordance, to national guidelines, all patients on anticoagulants should receive a standard yellow anticoagulant alert card. As a result of the audits, community pharmacy teams offered a standard yellow anticoagulant alert card to 24,576 patients from 2021–2022 and 26,427 patients from 2023–2024.

Fewer patients were prescribed concurrent antiplatelets with an anticoagulant (6021; 4.6% *p* < 0.001 in 2021–2022 to 4975; 4% *p* < 0.001 in 2023–2024.) Furthermore, for patients prescribed both anticoagulants and antiplatelets (which can put the patient at an increased risk of internal bleeding), the pharmacist contacted the prescriber 429 times in 2021–2022 and 309 times in 2023–2024, resulting in either the prescriber discontinuing the antiplatelet 33 times in 2021–2022 and 29 in 2023–2024 or gastroprotection being prescribed 217 in 2021–2022 and 151 in 2023–2024.

Although there was an increase in the number of patients prescribed NSAIDs with anticoagulants, more of these patients were also prescribed gastroprotection concurrently (927, 77.2% in 2021–2022 to 1457, 84.1%, *p* < 0.05). In patients who were prescribed NSAIDs concurrently with anticoagulants, there was an increase in the number of pharmacists contacting the prescriber to address safety issues and the risk of gastrointestinal bleeding. (1201 in 2021–2022 and 1732 in 2023–2024). Community pharmacy teams have also been proactive in identifying patients who are prescribed two anticoagulants simultaneously most likely as a result of switching to DOACs (132 in 2021–2022 and 184 in 2023–2024), which has resulted in several patients being discontinued on one or more anticoagulants as well as advice to return the discontinued anticoagulant back to the pharmacy for disposal (Table 7).

### 4.4. Study Limitations

Similar to related PQS studies, the data collected from everyday practice in England for the duration of both schemes has a high level of ecological validity [24,25,26]. Almost a quarter of a million patient entries were submitted, providing a snapshot of the clinical interventions and medicine optimisation activities undertaken by the community pharmacists and their team with respect to anticoagulants. The data were self-reported by community pharmacy teams without independent validation, so inconsistent data cannot be ruled out; however, most (111,195 84% in 2021–2022 and 104,677 83% in 2023–2024) of the data were included in the analysis. In addition, as per previous related studies, we acknowledge that pharmacy teams may behave differently when incentivised. The data relates to specific population group where data was collected over a three-year period and demonstrated an improvement in the activity to achieving the standards. This information could also be triangulated with the medication safety indicators, which are collected nationally and independent of this audit [5] Further studies could involve asking patients whether they report adverse events to medication in Pharmacovigilance systems. The data can include patients who presented with private prescriptions, although we anticipate this is likely to be a small number of patients. It is difficult to estimate the longer-term impact and sustainability of the recommendations from the community pharmacy national audits, but the outcomes from audit 2 would indicate that, by incorporating the recommendations from audit 1, improvements in pharmacy practice may have led to improvements in patient knowledge and access to standard yellow anticoagulant alert cards with a reduction in clinical interventions to optimise patient medication.

## 5. Conclusions

Community pharmacy teams are well placed in educating and counselling patients on the use of anticoagulants. Incentivising community pharmacies through the PQS to engage in audits has provided sufficient evidence that community pharmacists and their teams support the safe use of anticoagulants, reducing risks from drug interactions, side effects and switching to DOACs. This impact has supported the WHO’s medication safety challenge [27].

## Figures and Tables

**Figure 1 pharmacy-12-00134-f001:**
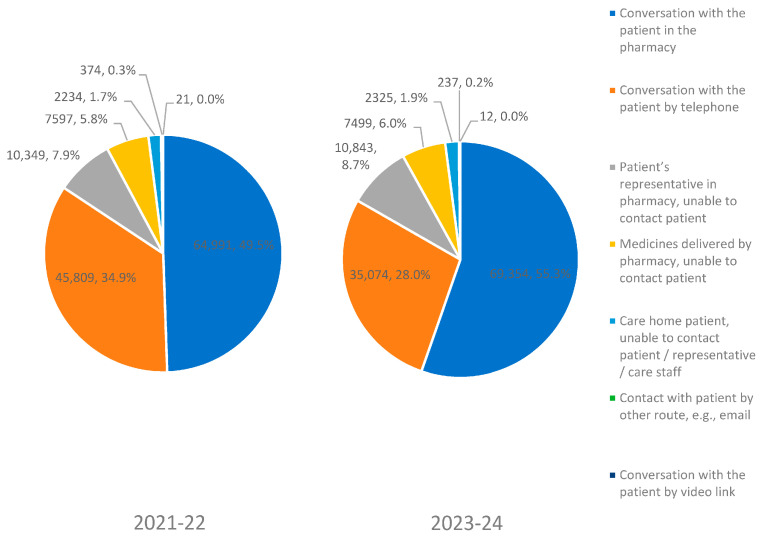
Methods of data collection.

**Table 1 pharmacy-12-00134-t001:** Audit results against audit standards.

	Audit Results		
Standard	2020–2021	2023–2024	*p* Value
**Standard 1: All patients are aware of or are provided with the following key information:** the medicine is an anticoagulant, i.e., a medicine to thin the blood/prevent blood clots; the symptoms of over anticoagulation, e.g., unexplained bruising, nose bleeds; to check with a doctor or pharmacist before taking OTC medicines, herbal products, or supplements. If taking a VKA, that dietary change can affect their anticoagulant medicine.	106,255 (95.6%) of patients knew they were prescribed an oral anticoagulant 85,029 (76.5%) of patients prescribed an oral anticoagulant knew the symptoms of anticoagulation 89,171 (80.2%) of patients were aware they need to check with a doctor or pharmacist before taking OTC medicines, herbal products, or supplements. 16,764 (67.4%) knew that dietary change can affect their anticoagulant medicine.	101,006 (96.5%) of patients knew they prescribed an oral anticoagulant 83,902 (80.2%) of patients prescribed an oral anticoagulant knew the symptoms of anticoagulation 85,071 (81.3%) of patients. were aware they need to check with a doctor or pharmacist before taking OTC medicines, herbal products, or supplement 12,594 (73.9%) knew that dietary change can affect their anticoagulant medicine	(*p* value < 0.001). (*p* value < 0.001). (*p* value < 0.001). (*p* value < 0.001).
**Standard 2: Alert Cards** All patients have a standard yellow anticoagulant alert card or are offered one.	73,901 (66.5%) of patients have a standard yellow anticoagulant alert card. 24,576 (96.5%) of those without a card were offered one.	76,735 (73.3%) of patients have a standard yellow anticoagulant alert card 26,427 (94.6%) of those without a card were offered one.	(*p* value < 0.001). (*p* value < 0.001).
**Standard 3: Safe use with other prescribed medicines—antiplatelets** Contact the prescriber about all patients prescribed an anticoagulant with an antiplatelet but not co-prescribed GI protection unless referral has been made in the last 6 months or the patient has already discussed with their prescriber.	6021 (4.6%) of patients were also prescribed an antiplatelet, of which 748 (12.4%) were not co-prescribed GI protection. Contact with the prescriber was made in 429 (57.4%) of these cases for review of the patient’s GI protection.	4975 (4.0%) of patients were also prescribed an antiplatelet of which 547 (11.0%) were not co-prescribed GI protection. Contact with the prescriber was made in 309 (56.5%) of these cases for review of the patient’s GI protection	(*p* value < 0.001), (*p* value < 0.05). (*p* value = 0.800)
**Standard 4: Safe use with other prescribed medicines—NSAIDs** The prescriber is contacted about all patients prescribed an anticoagulant with an NSAID.	0.9% (1201) of patients were concomitantly prescribed an NSAID. 927 (77.2%) of these patients were co-prescribed GI protection. 79.5% of patients taking anticoagulants and NSAIDs were referred to the prescriber.	1.4% (1732) of patients were concomitantly prescribed an NSAID. 1457 (84.1%) of these patients were co-prescribed GI protection. 80.5% of patients taking anticoagulants and NSAIDs were referred to the prescriber.	(*p* value < 0.001) (*p* value < 0.001) (*p* value = 0.576)
**Standard 5: INR monitoring and recording** INR monitoring within the last 12 weeks is confirmed for all patients prescribed VKAs.	18,446 (99.3%) of patients had INR monitoring within the last 12 weeks.	11,730 (99.1%) of patients had INR monitoring within the last 12 weeks.	(*p* value = 0.246)

*p* values of <0.05 or less indicate a statistically significant data result.

**Table 2 pharmacy-12-00134-t002:** Key Knowledge and Understanding.

Key Knowledge Understanding	Number of Patients (%) 2021–2022	Number of Patients (%) 2023–2024
**No. of patients already aware that they are taking an anticoagulant, i.e., a medicine to thin the blood/prevent blood clots:**		
Yes	106,255 (95.6)	101,006 (96.5)
No—information provided	4607 (4.1)	3478 (3.3)
No—information not provided	333 (0.3)	193 (0.2)
**No. of patients who already know the symptoms of over anticoagulation, e.g., unexplained bruising, nose bleeds:**		
Yes	85,029 (76.5)	83,902 (80.2)
No—information provided	25,478 (22.9)	20,308 (19.4)
No—information not provided	688 (0.6)	467 (0.4)
**No. of patients already aware of the need to check with the doctor or pharmacist before taking OTC medicines, herbal products, or supplements**		
Yes	89,171 (80.2)	85,071 (81.3)
No—information provided	21,033 (18.9)	18,889 (18.0)
No—information not provided	991 (0.9)	717 (0.7)
**Total**	**111,195 (100.0)**	**104,677 (100.0)**

**Table 3 pharmacy-12-00134-t003:** Patient knowledge, awareness, and the use of standard yellow anticoagulant alert cards.

Standard Yellow Anticoagulant Alert Card	Number of Patients (%) 2021–2022	Number of Patients (%) 2023–2024
**Patient has a standard yellow anticoagulant alert card**		
Yes, card not seen but patient confirmation they have the card	49,712 (44.7)	50,378 (48.1)
Yes, card seen by pharmacy staff	24,189 (21.8)	26,357 (25.2)
No card or unaware of card	25,459 (22.9)	27,942 (26.7)
Not known/Not reported	11,835 (10.6)	0 (0.0)
**Total**	**111,195 (100.0)**	**104,677 (100.0)**
**If no card or unaware of card, a standard yellow anticoagulant alert card offered to the patient**		
Yes, card accepted	12,914 (50.7)	12,018 (43.0)
Yes, but card declined because the patient has manufacturer’s alert card	8877 (34.9)	9461 (33.9)
Yes, but card declined because the patient has another anticoagulant alert card	1747 (6.9)	2519 (9.0)
Yes, but card declined for other reason	1038 (4.1)	2429 (8.7)
No, not offered	883 (3.5)	1515 (5.4)
**Total**	**25,459 (100.0)**	**27,942 (100.0)**

**Table 4 pharmacy-12-00134-t004:** Concomitant Prescribing: Antiplatelets.

Antiplatelet Co-Prescribed	Number of Patients (%) 2021–2022	Number of Patients (%) 2023–2024
Patients co-prescribed an antiplatelet	6021 (4.6)	4975 (4.0)
**Patient also prescribed gastroprotection:**		
Yes	5273 (87.6)	4428 (89.0)
No	748 (12.4)	547 (11.0)
**Total**	**6021 (100.0)**	**4975 (100.0)**
**Prescriber contacted for review of gastroprotection:**		
Yes	429 (57.4)	309 (56.5)
No	319 (42.6)	238 (43.5)
**Total**	**748 (100.0)**	**547 (100.0)**
**Prescriber contacted for review of GI protection:**		
Yes—prescriber discontinued one or both agents	33 (4.4)	29 (5.3)
Yes—prescriber confirmed no medication changes required	113 (15.1)	75 (13.7)
Yes—GI protection prescribed	217 (29.0)	151 (27.6)
Yes—other reason	66 (8.8)	54 (9.9)
No—prescriber has been contacted about GI protection for this patient within the last 6 months	89 (11.9)	56 (10.2)
No—patient has discussed with prescriber and has made decision not to take GI protection	153 (20.5)	124 (22.7)
No—other reason	77 (10.3)	58 (10.6)
**Total**	**748 (100.0)**	**547 (100.0)**

**Table 5 pharmacy-12-00134-t005:** Concomitant prescribing: NSAIDs.

NSAID Co-Prescribed	Number of Patients (%) 2021–2022	Number of Patients (%) 2023–2024
Patients co-prescribed an NSAID and anticoagulant	1201 (0.9)	1732 (1.4)
**Prescriber contacted about concomitant use of anticoagulant with NSAID:**		
Yes—prescriber discontinued one or both agents	151 (12.6)	190 (11.0)
Yes—prescriber confirmed both agents required	720 (60.0)	1006 (58.1)
Yes—other action by prescriber	84 (7.0)	132 (7.6)
No	246 (20.5)	339 (19.5)
Yes, gastro-protection prescribed	Not collected for audit 1	65 (3.8)
**Total**	**1201 (100.0)**	**1732 (100.0)**
**Patient also prescribed GI protection:**		
Yes	927 (77.2)	1457 (84.1)
No	274 (22.8)	275 (15.9)
**Total**	**1201 (100.0)**	**1732 (100.0)**

**Table 6 pharmacy-12-00134-t006:** Findings from VKA only.

		Number of Patients (%) 2021–2022	Number of Patients (%) 2023–2024
**Key knowledge**	**No. of patients already aware that dietary change can affect their anticoagulant medicine:**		
	Yes	16,764 (67.4)	12,594 (73.9)
	No—information not provided	353 (1.4)	207 (1.2)
	No—information provided	3868 (15.5)	2136 (12.5)
	Not applicable	3898 (15.7)	2106 (12.4)
	**Total**	**24,883 (100.0)**	**17,043 (100.0)**
**INR testing**	**For warfarin, INR test carried out**		
	4 to 12 weeks	5978 (32.2)	3731 (31.5)
	Less than 4 weeks	12,468 (67.1)	7999 (67.6)
	More than 12 weeks	138 (0.7)	103 (0.9)
	**Total**	**18,584 (100.0)**	**11,833 (100.0)**
**Actions taken**	**For patients, whose INR tests were more than 12 weeks ago, actions taken:**		
	Six monthly tests	2	0
	Advice given	1	0
	Advised patient to book test	42	40
	Contacted GP	8	20
	Contacted patient representation	2	0
	No action taken	9	8
	Patient to Contact GP	1	0
	Referred to GP	15	0
	Patient self-checks	3	0
	Stable patient	12	0
	Test booked	42	21
	Updated patient record	1	0
	Patient has different testing frequency	0	7
	Other	0	7
	**Total**	**138**	**103**

**Table 7 pharmacy-12-00134-t007:** Patients prescribed more than one anticoagulant and the action that resulted from pharmacist interventions.

More than One Anticoagulant Prescribed	Number of Patients (%) 2021–2022	Number of Patients (%) 2023–2024
**Patients prescribed more than one anticoagulant:**		
Patients on same drug but different strengths	99 (42.9)	0 (0.0)
Patients prescribed two different anticoagulant drugs	132 (57.1)	184 (100.0)
**Total**	**231 (100.0)**	**184 (100.0)**
**What action resulted from pharmacist intervention?**		
Referral to GP practice	41 (17.7)	47 (18.1)
Confirmed switching medication	86 (37.2)	43 (16.6)
Confirmed dose change (Advice given regarding dosing)	13 (5.6)	102 (39.4)
Confirmed on same drug but different strengths	77 (33.3)	0 (0.0)
use of dual therapy	0	2 (0.8)
No action/unknown outcome	10 (4.3)	3 (1.2)
Advice given regarding return of unwanted medication	0	50 (19.3)
Other	4 (1.7)	12 (4.6)
**Total**	**231 (100.0) ***	**259 (100.0) ***
**Actions resulted due to referral to GP Practice:**		
Confirmed switching medication	29 (70.7)	12 (25.6)
Unknown outcome	10 (24.4)	3 (6.4)
Confirmed dose change	1 (2.4)	0 (0.0)
Confirmed on same drug but different strengths	1 (2.4)	0 (0.0)
Prescriber continued both anticoagulants	0	7 (14.9)
Prescriber stopped one anticoagulant	0	24 (51.0)
Other	0	1 (2.1)
**Total**	**41 (100.0)**	**47 (100.0)**

* Multiple reasons could be ticked simultaneously.

## Data Availability

The data presented in this study are available upon reasonable request from the corresponding author.

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
