# Peer review of "An Evaluation of Oral Anticoagulant Safety Indicators by England’s Community Pharmacies"

_pharmacy, 2024, doi:10.3390/pharmacy12050134_

Round 1
Reviewer 1 Report
Comments and Suggestions for Authors
Dear Authors,
I had the pleasure of reading your work carefully. I believe the article is well-structured and well-written. The study is well-conducted. Here are some of my observations:
Introduction
The introduction effectively outlines the critical importance and potential risks associated with anticoagulant medications. However, I suggest adding a sentence at the end of the introduction to highlight the study's objectives.
Methods
The methods section is comprehensive and provides a broad overview of the study design, setting, participants, data collection, and analysis procedures.
· You mentioned that the data were collected using the NHS Business Services Authority's "snap survey", how does this tool work? It might be useful to add more details about this tool, and how the data is entered and managed. Moreover, does this explanation “The data collection tools included patient demographics; the name, dose and duration of anticoagulant prescribed; any concomitant medicines prescribed; whether gastroprotection was prescribed; conversations/contact with patients to support understanding of their medication; and whether the patient was referred to their general practitioner (GP) for a clinical review for suitable gastroprotection. A free-text option was available for general comments and additional information community pharmacy teams wanted to provide.” refer to the NHS Business Services Authority's snap survey?
Results
In general, the results section is comprehensive.
· Line 170: A total of 245,719 patients were audited, with 6605 community pharmacies participating in both audits and 10,899 pharmacies participating in either of the two audits. Did I understand correctly that the total number of participating pharmacies is 17,504?
· Line 178: Complete data were included in the final analysis. Some data entries contained blank data as representatives collected medication for patients, and they did not always know patient details.
For a more accurate analysis, did you exclude these incomplete data? If yes, please consider specifying it in the text for clarity. If not, please explain how you managed to include them.
· You provided the p-values, but there is no explanation of their significance. For a better understanding of the results, I would suggest briefly explaining the meaning of p-values in this context (for example, "A p-value of <0.001 indicates a statistically significant...").
Discussion
The discussion section effectively interprets and contextualizes the results of the audits.
· Line 273: By 2023-24, this had further decreased to 17,043 253 (16%) patients.
For clarity I suggest incorporating this sentence with the one at line 282. For example: In 2021-22, 24,545 patients (23%) were prescribed a VKA, decreasing to 17,043 patients (16%) by 2023-24, with the remaining patients prescribed DOACs.
· Line 282: these findings contrast with international data... Why do you think this difference exists between England and the international data? What could be a possible explanation in your opinion?

Author Response
Dear Reviewer 1,
thank you for your feedback. Please find our response in red attached.
thanks and best wishes

Reviewer 2 Report
Comments and Suggestions for Authors
- In the abstract, you should describe in more detail the methods used in the study.
- Lines 69 - 72 - You should add appropriate references.
- I recommend adding more information regarding the essential role of community pharmacists in ensuring safe anticoagulant use. Give examples of good practices from the UK or only England supported by appropriate references. Now there are only two sentences in the Introduction section describing the role of community pharmacists.
- Line 173 and Line 175 - Replace "See Figure. 1" with "(Figure 1)
- Line 197 - Replace "(See Table 2)" with "(Tabel 2)
Author Response
Dear Reviewer 2,
thank you for your feedback. Please find our response in red attached.
thanks and best wishes

Reviewer 3 Report
Comments and Suggestions for Authors
I read with interest the paper titled "Evaluation of Oral Anticoagulant Safety Indicators by England’s Community Pharmacies." The paper is well written and I have minor comments to improve the readiness of the paper.
1. Abstract - Please reduce the amount of text in the results and report a bit of discussion.
2. In the end of the introduction, please provide a clear aim for your study.
3. In the methods 2 audits were described, but then authors described 5 standards for the audits. Please clarify if more than one (or all) standards were applied in each audit. Audit questions could be shared as supplementary material, if not confidential.
4. Please check the verbal terms within the methods, as eg "All community pharmacy contractors in England are eligible...", maybe "were" instead of "are" is more appropriate.
5. Reccomendations from audit 1 is more a result than a methodology, right? I suggest to move to the results section and discuss accordingly.
6. Ethics section could appear only in the end, as already reported.
7. Figure 2 is not very informative (colors are the same) - I suggest to move to Supplementary material.
8. I suggest to check the statistical tests used. Chi-square tests could not be the best test to compare proportions in such populations. In the end, it seems strange that a small change in proportion of 0.9 from 95.6 to 96.5 (standard 1, topic 1) could have a p <0.001, whether a change of 1.0 from 79.5% to 80.5% (standard 4, topic 3) have a non-significant p-value, which of course is the expected. A parametric test to comparing both groups could be used instead, as t-test.
9. I wonder if there were any questions to assess the knowledge of the patients about the opportunity to report adverse events to such medication in Pharmacovigilance systems. Patients are well positioned to that, and pharmacy teams could also improve the patient experience on that. This could be a topic of discussion, if authors believe so.
Comments on the Quality of English Language
Proofreading of verbs in the methods should be done, to improve readiness.
Author Response
Dear Reviewer 3,
thank you for your feedback. Please see response in red attached.
thanks and best wishes
